# An Update on the Pathogenic Role of Neutrophils in Systemic Juvenile Idiopathic Arthritis and Adult-Onset Still’s Disease

**DOI:** 10.3390/ijms222313038

**Published:** 2021-12-02

**Authors:** Ji-Won Kim, Mi-Hyun Ahn, Ju-Yang Jung, Chang-Hee Suh, Hyoun-Ah Kim

**Affiliations:** Department of Rheumatology, Ajou University School of Medicine, Suwon 16499, Korea; jwk722@naver.com (J.-W.K.); mh2300@hanmail.net (M.-H.A.); serinne20@hanmail.net (J.-Y.J.); chsuh@ajou.ac.kr (C.-H.S.)

**Keywords:** neutrophil, neutrophil extracellular traps, systemic juvenile idiopathic arthritis, adult-onset Still’s disease, innate immune

## Abstract

Neutrophils are innate immune phagocytes that play a key role in immune defense against invading pathogens. The main offensive mechanisms of neutrophils are the phagocytosis of pathogens, release of granules, and production of cytokines. The formation of neutrophil extracellular traps (NETs) has been described as a novel defense mechanism in the literature. NETs are a network of fibers assembled from chromatin deoxyribonucleic acid, histones, and neutrophil granule proteins that have the ability to kill pathogens, while they can also cause toxic effects in hosts. Activated neutrophils with NET formation stimulate autoimmune responses related to a wide range of inflammatory autoimmune diseases by exposing autoantigens in susceptible individuals. The association between increased NET formation and autoimmunity was first reported in antineutrophil cytoplasmic antibody-related vasculitis, and the role of NETs in various diseases, including systemic lupus erythematosus, rheumatoid arthritis, and psoriasis, has since been elucidated in research. Herein, we discuss the mechanistic role of neutrophils, including NETs, in the pathogenesis of systemic juvenile idiopathic arthritis (SJIA) and adult-onset Still’s disease (AOSD), and provide their clinical values as biomarkers for monitoring and prognosis.

## 1. Introduction

Neutrophils are innate immune phagocytes with a short lifespan that arise from hematopoietic stem cells that account for 40% to 70% of circulating white blood cells in humans [1]. As the first line of defense in the innate immune system, they play a key role in eliminating invading pathogens [2]. The main offensive mechanisms of neutrophils are phagocytosis, release of granules, and production of reactive oxygen species (ROS). Furthermore, the formation of neutrophil extracellular traps (NETs) has been described as a novel defense mechanism in the literature [3]. NETs are a network of fibers assembled from chromatin deoxyribonucleic acid (DNA), histones, serine proteases, cytoskeletal proteins, and antimicrobial peptides, which are formed and released from the extracellular space during the elimination of pathogens that have entered the body [4,5,6]. With the discovery of NETs, shifting the perception that neutrophils are far more heterogeneous than initially estimated, it is becoming clear that these cells have functional aspects beyond mere immune defense [7]. In addition to NET formation, neutrophils exhibit a wide range of phenotypes according to nuclear morphology, migratory and phagocytic capacity, and immunomodulatory functions, reflecting variability from the developmental stage to the activation status. The heterogeneity of the rapidly changing phenotypic and functional properties of neutrophils under pathophysiological conditions represents an opportunity to intervene in a variety of diseases, including vascular and immune-related diseases [8].

Currently, increasing evidence shows that neutrophil heterogeneity and NETosis could be strong inducers of autoimmunity [9]. In terms of neutrophil heterogeneity, one possible explanation might be that putative pathogenic neutrophil subsets are a key aspect in the development of autoimmune responses and the most widely studied in the context of autoimmunity is low-density granulocytes (LDGs) [10]. Moreover, the composition of NETs may vary depending on the stimulus, which means that there is a disease associated with NETs [11]. The association between increased disease-specific NETs and autoimmunity was first reported in antineutrophil cytoplasmic antibody-related vasculitis, and the role of NETs in various diseases, including systemic lupus erythematosus, rheumatoid arthritis (RA), and psoriasis, has since been elucidated in the literature [12,13,14,15]. Compared to studies on autoimmune diseases, relatively few studies have shown that activated neutrophils and NETosis contribute to pathogenicity in rare systemic autoinflammatory diseases such as systemic juvenile idiopathic arthritis (SJIA) and adult-onset Still’s disease (AOSD).

Autoinflammatory diseases are a group of rare diseases characterized by seemingly unexplained fever and inflammation affecting multiple organs, which belong to an area of clinical diseases other than autoimmune diseases [16]. Both SJIA and AOSD are autoinflammatory diseases with similar clinical features and laboratory findings, with a high spiking fever, skin rash, generalized lymphadenopathy, hepatosplenomegaly, and leukocytosis [17,18]. Since AOSD is known as the manifestation of SJIA in adults, SJIA and AOSD are two expressions of the same disease in different hosts [19]. Diagnosis of SJIA and AOSD is difficult because it is essential to rule out similar conditions such as infection or malignancy, and it uses criteria based on clinical evaluation without specific tests or laboratory results [20,21]. Several biomarkers, including acute phase reactants, inflammatory cytokines, and chemokines, are intended to be used for diagnosis in routine practice; however, their clinical utility is still unclear and nonspecific [22]. The sharing of several features implies that similar pathogenic mechanisms may underlie in the development of these two diseases. Unprovoked activation of the innate immune system and overproduction of proinflammatory cytokines play a pivotal role in disease pathogenesis, while the role of adaptive immunity is limited [23,24,25,26].

Despite thousands of reports available on SJIA and AOSD, our understanding of the exact pathogenesis is not yet complete. New research results on how neutrophil activation and NETosis cause amplification of inflammation in SJIA and AOSD are also gradually being provided, although the progress is slower than it is for other autoimmune and autoinflammatory disorders [27,28]. A better understanding of the role of neutrophils and NETs in the pathogenesis of SJIA and AOSD may lead to the discovery of new biomarkers. Herein, we discuss the mechanistic role of neutrophils and NETs in the development and progression of SJIA and AOSD, and provide their clinical values as biomarkers for monitoring and prognosis.

## 2. Neutrophils and NETs in Sterile Inflammation

Neutrophils are the first cells recruited at an inflammatory site during invasion of microorganisms or tissue damage, and they seem to orchestrate both the role of acting against pathogens and facilitating resolution of inflammation [29]. In the early stages of inflammation, the initiating stimulus is sensed by pathogen-associated molecular patterns (PAMPs) or damage-associated molecular patterns (DAMPs), leading to the release of pro-inflammatory mediators, including lipid mediators such as prostaglandins, leukotriene B4, and complement anaphylatoxins C3a and C5a along with many cytokines and chemokines (e.g., tumor necrosis factor (TNF)-*α*, interleukin–1 (IL–1), and CXC chemokines) [30]. Multiple pro-inflammatory mediators act as potent chemoattractants to alert innate immunity and contribute to neutrophil recruitment [31].

Pro-inflammatory mediators attract monocytes and macrophages in addition to neutrophils to the site of inflammation to eliminate the stimulus [32]. Simultaneously, with the recruitment of these phagocytic cells, a switch in lipid mediator type occurs, resulting in the biosynthesis of pro-inflammatory lipid mediators into anti-inflammatory lipid mediators, such as lipoxins, resolvins, and protectins, to attenuate inflammation [33]. These novel mediators initiate the resolution of inflammation, in which neutrophils restrict further infiltration and undergo apoptosis, and this cellular debris generated during apoptosis is removed by macrophages [34]. Through this series of processes, neutrophils must be cleared before they cause some subsidiary damage to host tissues and unwanted inflammation in the inflamed tissues, which is essential for the maintenance of normal tissue homeostasis [35].

The traditional view of neutrophils as simple scavengers of pathogens has radically expanded to new perspectives, as the unrecognized complexity of neutrophils is uncovered in research [36]. Until 2004, when NETs were first reported, phagocytosis, ROS production, and protease activity were thought to be the major neutrophil effector mechanisms [37]. NETs have brought neutrophils back to the spotlight of immunological research, as they are at the forefront of new interest due to their broadly effective pathogen clearance. NETosis is a program for the formation of NETs, which begins with activation of neutrophils through the recognition of stimuli, and the process is divided into lytic NETosis and non-lytic NETosis. [38]. In lytic NETosis, nicotinamide adenine dinucleotide phosphate (NADPH) oxidase is activated via the Raf-MEK-ERK signaling pathway, thereby generating ROS and activating protein-arginine deiminase type 4, inducing NET formation through chromatin decondensation [39]. NETosis ends with the release of chromatin from neutrophils along with associated neutrophil proteins, resulting in neutrophil cell death. It is also referred to as suicidal NETosis, and this process takes approximately 2–4 h [40]. Non-lytic NETosis, called vital NETosis, releases NETs within 5–60 min without cell death via Toll-like receptor (TLR) 2, TLR4, or complement receptors independent of NADPH oxidase activation [41].

Initially, neutrophils and NETs were considered to be involved in antimicrobials, while a number of studies had shown that they also played a pivotal role in the development of sterile inflammation under several chronic inflammatory conditions [42]. Sterile inflammation occurs in the absence of microorganisms, and it can be induced by physical, chemical, metabolic, or antigenic stimuli. When these non-infectious stimuli persist and cannot be eliminated, they promote sterile inflammation, becoming a part of the pathophysiology of numerous human diseases, such as metabolic disorders, autoimmune diseases, autoinflammatory diseases, and cancer [43,44,45,46]. Delayed neutrophil apoptosis or prolonged NET formation is the root cause of inflammatory disorders [47,48].

## 3. Neutrophils and NETs in Autoinflammatory Diseases

The term “autoinflammatory disease” was first coined in 1999 to represent an emerging family of clinical disorders that are different from autoimmune diseases [49]. Autoinflammatory and autoimmune diseases have similarities in that self-tissue-directed inflammation occurs without obvious infectious triggers or injuries; however, the two categories can be distinguished by pathways of innate or adaptive immune responses [50]. Autoinflammatory diseases are characterized by episodes of recurrent unprovoked inflammation that cause aberrant activation of the innate immune system without autoantibodies or auto-reactive antigen-specific T cells [51]. While most of these have strong genetic factors with mutations in a single gene, there are also polygenic or multifactorial origins without identified genetic mutations, such as SJIA, AOSD, and inflammatory bowel disease (IBD). More than 30 new genes associated with autoinflammatory diseases have been identified since the mutation of Mediterranean fever (MEFV) gene was first discovered as the cause of familial Mediterranean fever (FMF) [52,53,54]. The important pathogenic mechanisms have been observed to be driven by mutations in genes involving inflammasome activation or cytokine pathways, including NOD-like receptor pyrin domain-containing protein 3, TNF receptor 1, and IL-1 receptor antagonist [55].

As neutrophils are the key cells explaining the development of autoinflammation, it has been significantly elucidated that NETs contribute to certain aspects of the pathogenesis of systemic autoinflammatory diseases. Several autoinflammatory diseases mediated by a sustained influx of neutrophils and NET release and their proposed roles are discussed accordingly in the literature.

### 3.1. Familial Mediterranean Fever

FMF is the most common hereditary autoinflammatory disorder, and it mainly affects individuals in the eastern Mediterranean region. Clinically, it is characterized by recurrent, self-limited episodes of fever caused by neutrophil-induced serosal inflammation, including synovitis, pleuritis, and pericarditis [56,57]. FMF is associated with mutations in the MEFV gene encoding the protein pyrin, which is highly expressed primarily in neutrophils. Neutrophils are the main effector cells of acute inflammatory attacks in patients with FMF, and disease attacks result in neutrophilia and a massive influx of neutrophils into the inflamed sites [58,59]. Mutated pyrin reduces autophagic function by impairing its role as a selective autophagy receptor targeting inflammasome components, leading to the release of IL–1*β*, a key cytokine in this disease [60,61,62]. Recent experimental evidence suggests that neutrophils release large amounts of NETs in an autophagy-dependent manner during FMF inflammation. These NETs are transmitted to extracellular space bioactive IL–1*β*, which amplifies IL–1*β* release during the first 24 h of acute inflammation attacks. At the same time, FMF inflammation is suppressed by a negative feedback mechanism of NETs that interferes with IL–1*β* activity by dismantling the NET chromatin scaffold by DNase. Compensatively, the autophagy of neutrophils is lowered, thereby attenuating the release of NETs and protecting them from crisis [63]. In addition, regulated in development and DNA damage response 1 (REDD1), a modulator of neutrophil function upstream of pyrin, is involved in the release of NETs and regulation of IL–1*β*, and it has emerged as a novel link in the mechanisms leading to FMF attacks [64].

### 3.2. Gout

Gout is classified as an autoinflammatory disease that is caused by monosodium urate (MSU) crystals, which act as DAMPs and can contribute to the initiation of acute joint inflammation by activating the innate immune system [65]. Uptake of MSU crystals through phagocytosis induces the activation of NLRP3 inflammasome and release of IL–1*β*, which not only promotes the recruitment of neutrophils and other immune cells, but also secretes more inflammatory mediators such as IL–6, IL–8, and TNF–*α* [66,67,68]. Early studies of gout and NETs revealed that NETs were induced in synovial cells and peripheral neutrophils during the acute phase of gouty inflammation to assemble pro-inflammatory responses, whereas recent studies have shown that aggregated NETs can inhibit inflammatory responses through the degradation of inflammatory cytokines and chemokines [69,70,71]. In this process, acute gout inflammation is naturally resolved within a few days, leaving the affected tissue with an abscess-like creamy lump containing MSU crystals, dead immune cells, and NETs, defined as tophi [72]. Although tophi formation is a mechanism to resolve inflammation, it accumulates in joints and parts of the body for long periods of time, contributing to chronic gout inflammation, leading to the destruction of surrounding tissue [71,73].

### 3.3. Inflammatory Bowel Diseases

Crohn’s disease (CD) and ulcerative colitis (UC), the most widely known IBD, are a group of relapsing and chronic diseases that cause damage to the gastrointestinal tract due to a dysregulated immune response to unknown stimuli [74]. Neutrophils, regarded as first responders to inflammation, are important entities that influence the gut mucosal inflammatory milieu in IBD [75]. In the early stages of intestinal inflammation in patients with IBD, neutrophils flow into the intestinal mucosa and recognize and phagocytose pathogens to heal the mucous membrane. If this process is not tightly regulated, a vicious cycle of self-amplification loop occurs in which a large number of neutrophils accumulate in the inflamed mucous membrane, producing excessive ROS and proinflammatory cytokines, thereby maintaining persistent intestinal inflammation and eventually impairing intestinal barrier function [76,77,78]. Several studies have demonstrated an increased presence of NETs in inflamed gut mucosa, stool, or blood in patients with IBD, and there have also been reports that the abundance of NETs is positively correlated with disease activity [79,80,81]. Among IBD, the response of neutrophils to autoinflammatory properties, such as REDD1, autophagy, and the NETosis pathway, is strongly featured, especially in UC. Studies have reported that ROS generation is enhanced in CD, whereas neutrophil migration is reduced, phagocytic functions are impaired, and evidence of NET formation is scarce therein [79,82,83]. This suggests that the activity of neutrophils is also different in the two diseases as seen by clinical symptoms, and the location and layer affected in the gut are different.

## 4. Neutrophils in SJIA and AOSD

Neutrophil activation, a hallmark of the pathogenesis of SJIA and AOSD, is responsible for the initiation and development of inflammation by releasing a wide variety of granular enzymes and antimicrobial proteins. In an acute flare of the disease, more than 80% of patients have neutrophilic leukocytosis, which can be a differentiating feature of SJIA and AOSD from other rheumatic diseases. Several lines of evidence indicate the crucial role of various activating cell surface receptors and intracellular signaling mechanisms expressed in neutrophils involved in the pathogenesis of SJIA and AOSD (Figure 1). Studies related to the role of neutrophils in SJIA and AOSD are summarized in Table 1.

Initially, levels of C–X–C motif chemokine ligand (CXCL)–8, known to be critical for neutrophil recruitment and activation, significantly increase during the inflammatory response associated with AOSD [84]. Similarly, Kasama et al. detected that CX3CL1, whose primary role is to promote neutrophil binding, adhesion, and activation of target cells, is positively correlated with disease activity of AOSD and that high CX3CL1 reflects the presence of hemophagocytic syndrome [85]. Another molecule facilitating adhesion and retention of neutrophils at the inflamed sites, intracellular adhesion molecule 1, and neutrophil activation marker CD64 (Fc*γ*R1) were upregulated in active untreated AOSD patients, and these were correlated with disease activity scores [86,87]. C–type lectin domain family 5–member A (CLEC5A), which combines with adapter proteins to form receptor complexes and which is involved in inflammatory responses through differentiation and activation of neutrophils, has been demonstrated to be overexpressed in AOSD patients. A positive correlation between CLEC5A levels and inflammatory parameters or disease activity of AOSD has also been reported in the literature [88]. The frequencies of circulating CD11b, the molecules of complement receptor 3, and CD32, most widely expressed in Fc*γ*R, were observed in whole blood cells of patients with AOSD than in patients with RA or healthy controls [89]. CD11b and CD32 are mainly expressed in neutrophils, suggesting that neutrophils contribute significantly to the onset of AOSD. Other receptors related to neutrophil activation, such as granulocyte-macrophage colony-stimulating factor receptor, are being mentioned as targets for new therapies for AOSD; however, evidence for their pathological roles in AOSD is still lacking [98].

LDGs, known as the key drivers of inflammation in various autoimmune diseases, cancer, and sepsis, are separated by density from normal density granulocytes [99]. LDGs with pro-inflammatory features were also found to be increased in active SJIA and AOSD. Increased IL-6 producing LDGs contributes to the pathophysiology of AOSD, and activation of neutrophils due to elevated transcription of genes encoding LDGs with pro-inflammatory features has potential implications for the mechanism of SJIA [90,91]. Additionally, evidence of functional and gene expression of neutrophil alterations has been reported in SJIA patients, and persistent proinflammatory activation in neutrophils has been shown not only in active disease but also in long-standing clinically inactive disease in SJIA [90,92].

Significantly higher levels of phagocyte-secreted pro-inflammatory cytokines IL–1, IL–6, and IL–18 in patients with SJIA and AOSD compared to all other inflammatory diseases can also support the role of neutrophils [93,94,95]. Moreover, a large amount of data on the elevation of various molecules expressed and secreted by neutrophils in response to inflammatory stimuli suggest the importance of neutrophils in the pathogenesis of SJIA and AOSD. Chemokines, including CXCL–9, 10, 11, 12, 13, and CXC receptor 4, were found in high concentrations in the serum of patients with SJIA and AOSD [94,96,97].

## 5. NETs in SJIA and AOSD

Recently, growing evidence has suggested that the release of NETs is explained by the mechanisms of SJIA and AOSD (Table 2). Figure 2 shows the mechanisms of NET involvement in SJIA and AOSD. The main protein components of NETs are histones, followed by granule-derived peptides and enzymes such as neutrophilic elastase, myeloperoxidase, calprotectin, cathepsin G, leukocyte proteinase 3, lysozyme C, and neutrophil defensins [100]. In SJIA and AOSD, the most frequently investigated field related to NETs is the S100 protein. The term calprotectin is described as a complex of two calcium-binding proteins of the S100 family of proteins, S100A8 and S100A9 [101]. In addition, S100A8/S100A9 complex, S100A8, S100A9, and S100A12 belonging to the S100 family are released from neutrophils as part of NETs and they function as DAMPs [102]. The S100 family of proteins, which are mainly derived from neutrophils and macrophages, are significantly higher in patients with SJIA and AOSD than in healthy controls. They act as ligands of TLR4 or receptor for advanced glycation end products (RAGE) to accelerate neutrophils and activate the release of pro-inflammatory cytokines, contributing to systemic inflammation of AOSD through enhancement of the feedback loop [103]. Serum levels of S100A8/A9 and S100A12 are significantly higher in patients with SJIA and AOSD, and they are strongly correlated with peripheral blood neutrophil levels [92,104,105,106]. Serum S100A8/A9 levels have been proven to have a positive correlation with inflammatory markers of AOSD, and they are associated with neutrophil infiltration in the skin and lymph nodes affected by AOSD. Serum S100A12 was also found to be strongly correlated with disease activity markers of SJIA and AOSD [105,106,107].

The levels of cell–free DNA, *α*–defensin, and MPO–DNA complex, which are remnants of NETs, have been found to increase significantly in the serum of patients with AOSD compared to healthy controls. These NET molecules were associated with skin rash and lymphadenopathy, indicating the induction of IL–1*β* production in monocytes as a pathogenesis of AOSD [27]. In another study, the levels of cell–free DNA and NET–DNA complexes were significantly increased in patients with AOSD compared with healthy controls, and NET formation accelerated pathogenic mechanisms through activation of NLRP3 inflammasomes and macrophages for pro-inflammatory cytokine release [28]. Moreover, urinary proteins enriched in neutrophil degranulation and neutrophil-derived lipocalin–2 were highly expressed in patients with AOSD [108,109].

A previous study demonstrated that IL–18, a pivotal cytokine of AOSD, induces NET by enhancing calcium influx into neutrophils. Another molecule involved in this process is microRNA–223, which modulates the immune response of the host. In this study, a fine-tuned mechanism was proposed in which IL–18, microRNA–223, and NET formation regulate inflammation in AOSD, in which NET formation inhibits calcium influx into neutrophils by upregulation of microRNA–223, and it eventually inhibits IL–18 mediated NET formation [110]. The role of LDGs and NET formation in AOSD was also identified, and it has been reported that the concentration of high mobility group box–1 protein (HMGB–1) and cathelicidin LL–37 in NETs was increased in patients with AOSD [111]. HMGB–1 is a nuclear protein released from various cells and is a DAMP that interacts with TLR4, TLR2, and RAGE to act as a major mediator of the immune response [118,119]. Serum HMGB–1 level was the most elevated at the time of diagnosis in patients with AOSD and SJIA, and most of the follow-up patients showed decreased levels after the disease showed an improvement. In patients with AOSD, serum HMGB–1 may be correlated with clinical disease activity, such as C–reactive protein (CRP), systemic score, skin rash, and sore throat [112,113,114]. LL–37 is released in the form of NETs via TLR 8/13 and formyl peptide receptor-like 1 in neutrophils, inducing the production of inflammatory cytokines, leading to the generation of interferon-α by dendritic cells. A positive loop was identified in which LL–37 activates the NLRP3 inflammasome and releases IL–1*β* and IL–18 through caspase-1, which stimulates NET release [120]. Differential expression of LL–37 was not found according to the disease activity status of AOSD [111]. In addition, circulating NETs have the potential to discriminate the involvement of the hepatic and cardiopulmonary systems among patients with AOSD, and they may help to provide therapeutic strategies by predicting their glucocorticoid response [115].

Recently, there has been a report of genetic susceptibility to neutrophil activation and NETosis in AOSD. Evidence has emerged that *LILR3*, a gene of the leukocyte immunoglobulin–like receptor (LIR)–A3, plays a pathogenic role in AOSD. Plasma levels of LIR–A3 were elevated in patients with *LILRA3* ^+/+^, and they correlated with disease activity measures and circulating NET–DNA complex levels [116]. Although various roles of NETs have been identified in AOSD, data on the role of NETs in SJIA are still limited. In one study, Hu et al. demonstrated that active SJIA patients release a large amount of histones from NETs and that induced-serum histones promote the propagation of NETs, resulting in malignant feedback mechanisms, which may be the pathogenesis of SJIA [117]. A recent study showed that secondary neutrophil granule proteins, such as human neutrophil lipocalin and myeloperoxidase, were significantly increased in patients with SJIA [108]. Currently, many studies are underway to elucidate the role of NETs in the pathogenesis of SJIA; thus, a clearer mechanism is expected to be identified in the future.

## 6. Clinical Values as Biomarkers Related to Neutrophils for Monitoring and Prognosis in SJIA and AOSD

Several studies have evaluated the role of cellular or serum markers related to neutrophils as biomarkers of SJIA and AOSD. A biomarker is defined as an objective measurement of a molecular or biological event, whose changes can reflect disease pathogenesis [121]. Therefore, several markers related to neutrophils and NETs could be good biomarkers for the diagnosis and evaluation of disease activities in SJIA and AOSD. In this section, we present a review of the potential use of markers related to neutrophils for the diagnosis, evaluation of disease activities, and prognosis in SJIA and AOSD patients.

Several proinflammatory cytokines and chemokines related to neutrophil recruitment or activation and NET have been evaluated in the serum of patients with SJIA and AOSD [18,95,96,97,122,123,124]. Significantly higher levels of IL–6, CXCL8, and IL–18 in sera or plasma were found in patients with active untreated AOSD or SJIA than in healthy controls [123,124,125]. Furthermore, the mRNA expression levels of IL–18 and CXCL8 were significantly higher in the biopsy tissue of Still’s rash and synovial membranes of AOSD patients compared with those in controls [124]. Interestingly, AOSD patients with chronic articular patterns had significantly higher levels of serum CXCL8 than those with a monocyclic systemic pattern [124]. Therefore, CXCL8 could be a good marker for predicting chronic articular patterns in AOSD. Several chemokines and receptors, including CXCL9, 10, 11, 12, 13, and CXCR3 and CXCR4, were found in high concentrations in the serum of patients with SJIA or AOSD, and these correlated with several disease activity markers of SJIA or AOSD [96,97,126,127]. In particular, the levels of CXCR9, CXCL10, CXCL11, and IL–18 were elevated in SJIA patients with macrophage activation syndrome compared with active SJIA without macrophage activation syndrome [125,126]. Therefore, interferon-γ-related chemokines and IL–18 could predict poor prognosis related to macrophage activation syndrome in patients with SJIA. A recent study aimed to identify biomarkers in a prospective cohort of patients with bloodstream infection and AOSD for differential diagnosis, and found that a combination of plasma IL–18 and ferritin levels could be used to distinguish bloodstream infections from AOSD (sensitivity: 96.15% and specificity: 100%) [128].

S100 proteins, such as S100A8/A9 and S100A12, have become reliable biomarkers for differential diagnosis, evaluation of current disease activity, and prediction of further disease flares in SJIA or AOSD [104,105,106,107]. Serum S100A8/A9 levels were elevated in active AOSD or SJIA, and they correlated with disease activity markers, such as CRP, ferritin, and systemic scores. S100 protein was also suggested to be a predictive marker of further disease flares during clinical remission in patients with SJIA [104,105,129]. Furthermore, S100A8/A9 was superior to CRP in differentiating systemic JIA from other fever syndromes, such as systemic undifferentiated recurring fever syndrome [130]. In a study with AOSD, the sensitivity and specificity of S100A8/A9 using receiver operating characteristic curves for the diagnosis of AOSD from other rheumatic diseases including RA, OA, and SLE was evaluated, and a sensitivity of 63% and a specificity of 80% were reported accordingly [131]. Studies on the role of S100A12 as a biomarker were similar to those on S100A8/A9 in SJIA and AOSD patients, with these levels being elevated in active SJIA or AOSD patients as compared to HC and associated with inflammatory markers, such as ESR, CRP, and ferritin, and the systemic score [106,107]. Furthermore, S100A12 levels were elevated in patients with SJIA or FMF compared to those in patients with other fever diseases, such as acute myeloblastic leukemia, acute lymphoblastic leukemia, or systemic infection, and their sensitivity and specificity to distinguish between SJIA and infections were 66% and 94%, respectively [106]. Another study showed that S100A12 levels were increased in patients with SJIA, compared to systemic undifferentiated recurring fever syndrome, other defined autoinflammatory syndromes, and non-systemic JIA. The area under the receiver operating characteristic curve (AUC = 0.9) of S100 proteins for differentiating SJIA from other fever syndromes was superior to that of CRP (AUC = 0.7) [130]. Therefore, these data suggest that serum S100 proteins, such as S100A8/A9 and S100A12, are good biomarkers for the differential diagnosis of systemic JIA and AOSD, and for the evaluation of disease activity.

Several recent studies showed elevated cell-free DNA and NET–related molecules in patients with active AOSD or SJIA, compared to those with inactive status or HC [27,28,63,109,117]. Furthermore, these levels, such as MPO–DNA and *α*–defensin, were correlated with the levels of disease activity markers of AOSD. In a recent study, patients with active SJIA had more extracellular histones released by activated neutrophils than those with inactive SJIA and HC. They also correlated significantly with clinical disease activity [117]. However, these studies did not evaluate the associations between NET molecules and the prognosis of AOSD or SJIA. Therefore, further studies are needed to evaluate the role of NET molecules as prognostic markers in patients with AOSD or SJIA.

## 7. Conclusions

This review describes the roles of neutrophils and their release of NETs, as DAMPs, that contribute to aggravation of inflammation in the pathogenesis of AOSD and SJIA. Although the initial trigger of inflammation in SJIA or AOSD is not well known, viral or bacterial infection and environmental trigger factors as danger signals could lead to the activation of neutrophils, leading to the transmigration into inflamed tissues and the production of proinflammatory cytokines and chemokines. Several specific receptors on neutrophils, including pattern-recognition receptors and Fc receptors interacting with danger signals, further increase neutrophil recruitment and activation. Neutrophil activation and mediator release form a positive-feedback loop that enhances neutrophil recruitment and amplifies inflammatory responses, which may act as an important axis of pathogenesis in AOSD and SJIA. In addition, activated neutrophils produce and release NETs, proinflammatory cytokines, and chemokines and activate and interact with macrophages in the inflamed tissue. Furthermore, several DAMPs, such as HMGB–1, S100 proteins, LL37, and MPO–DNA complex, are elevated in the blood of patients with SJIA or AOSD, accelerating systemic inflammation with proinflammatory cascades through several pattern recognition receptors of neutrophils.

Despite the continuous research conducted so far on this subject, undoubtedly, further research in this regard will provide greater clarity in understanding the pathogenesis of AOSD and SJIA. The first question is whether the initial danger signal for activation of neutrophils is truly provoked, or whether increased danger signaling is due to a loss of inhibitory signal in neutrophil activation during the initial inflammatory stages of AOSD. The second is whether neutrophil activation and NETs are associated with chronic articular forms of AOSD, while they are thought to contribute clearly to the initial acute inflammation of AOSD. The third is how much of the inflammation of SJIA and AOSD is caused by neutrophils, and how effective it would be when it is blocked. Ongoing research into the role of neutrophils and NETs in the pathogenesis of SJIA and AOSD will provide a better understanding of disease pathogenesis.

The current findings enhance our understanding of the pathogenesis of systemic JIA and AOSD, and they will facilitate the development of novel therapeutics targeting neutrophils and NET molecules and the discovery of biomarkers related to neutrophils that could be used to diagnose and predict prognosis in systemic JIA and AOSD.

## Figures and Tables

**Figure 1 ijms-22-13038-f001:**
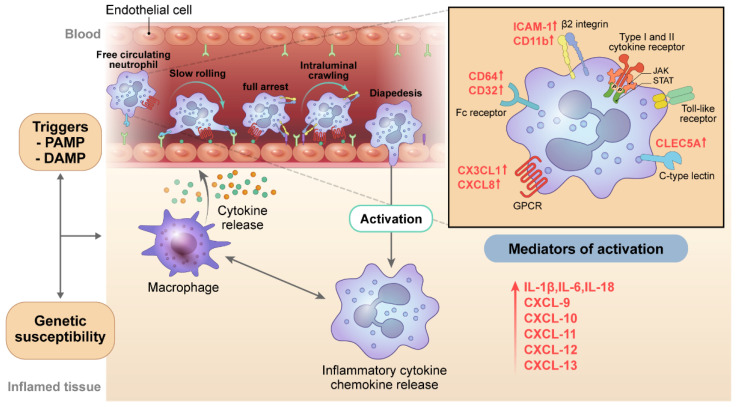
Overview of the role of neutrophils in AOSD and SJIA. Circulating neutrophils are triggered by the release of pathogen-associated molecular pattern molecules (PAMPs) or damage-associated molecular pattern molecules (DAMPs) during inflammation and transmigrate into inflamed tissues. In AOSD and SJIA, a variety of specific receptors on neutrophils, including pattern-recognition receptors and Fc receptors interacting with PAMPs and DAMPs increase, further promoting neutrophil recruitment and activation. Activated neutrophils release more cytokines/chemokines and communicate with macrophages in the innate immune system. Neutrophil activation and mediator release form a positive-feedback loop that enhances neutrophil recruitment and amplifies inflammatory responses, acting as an axis of pathogenesis in AOSD and SJIA.

**Figure 2 ijms-22-13038-f002:**
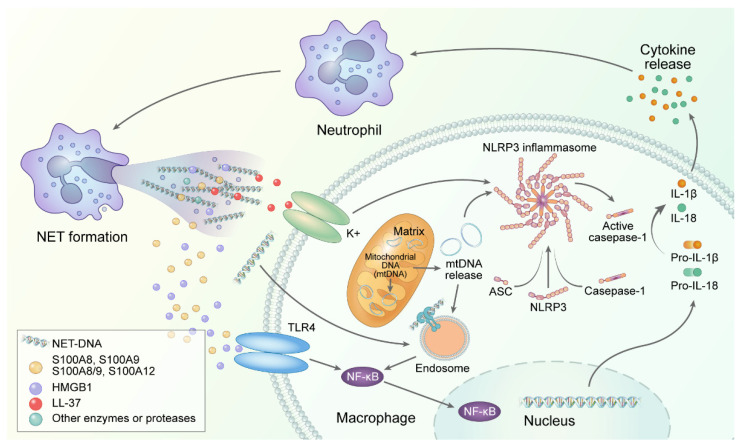
Neutrophil extracellular traps and its implications in AOSD and SJIA. Several neutrophil extracellular traps (NETs) released through neutrophils such as S100 protein, high mobility group box-1 protein, and LL–37 accelerate inflammation with a proinflammatory cascade through toll–like receptor 4 or receptor for advanced glycation end products or inflammasome activation. NET formation promotes cytokine storms by linking neutrophils and macrophages.

**Table 1 ijms-22-13038-t001:** List of studies indicating association between neutrophils and SJIA/AOSD.

Study Population	Main Findings	Country	Author, Year [ref.]
Analysis of serum from 14 patients with AOSD	Overproduction of CXCL–8 may contribute to the pathogenic mechanism of AOSD.	South Korea	Choi J-H et al., 2003 [84]
Analysis of serum from 19 patients with AOSD and 19 HCs	Serum CX3CL1 level may be used as a clinical marker to assess the disease activity of AOSD, and high serum CXCL–8 and ferritin reflected the presence of hemophagocytic syndrome.	Japan	Kasama T et al., 2012 [85]
Analysis of serum from 50 patients with untreated AOSD, 20 with RA, and 20 HCs	Serum levels of ICAM–1 were significantly elevated in patients with active untreated AOSD compared with those with active RA and HCs.	Taiwan	Chen DY et al., 2005 [86]
Analysis of serum from 10 patients with AOSD	Neutrophil CD64 is upregulated in patients with active AOSD	Japan	Komiya A et al., 2012 [87]
Analysis of serum from 34 patients with AOSD and 12 HCs	C-type lectin domain family 5–member A is involved in the pathogenesis and may serve as an activity indicator of AOSD.	Taiwan	Chen P-K et al., 2020 [88]
Analysis of serum from 13 patients with AOSD, 19 with RA, and 19 HCs	Significantly higher frequencies of cells presenting CD11b and CD32 from whole blood cells in patients with AOSD than in patients with RA or HC	Korea	Kim HA et al., 2017 [89]
Analysis of serum from 16 patients with JIA (3 SJIA) and 19 HCs	Neutrophils from JIA patients have elevated transcription of genes encoding granule proteins, a major cause of inflammation.	UK	Ramanathan K et al., 2018 [90]
Analysis of serum from 56 patients with AOSD (active 32 and inactive 24) and 26 HCs	Active AOSD is associated with elevated levels of low-density granulocytes that produce IL–6	China	Liu Y et al., 2021 [91]
Analysis of serum from 23 patients with active SJIA and 22 with inactive SJIA	Neutrophil activations, S100 alarmin release, and proinflammatory gene expression were seen in SJIA patients with both active disease and clinically inactive disease.	United States	Brown RA et al., 2018 [92]
Analysis of serum from 39 patients with SJIA (active 25 and inactive 14) and 17 HCs	IL-6 plays a significant role in the pathogenesis of SJIA.	Italy	De Benedetti F et al., 1991 [93]
Analysis of serum and synovial fluid from 65 patients with JIA (20 SJIA), 9 with type I diabetes, and 20 HCs	Several cytokines including IL18 may correspond to the activation status during inflammation in JIA	The Netherlands	De Jager W et al., 2007 [94]
Analysis of serum from 23 patients with SJIA, and 12 HCs	IL-1 is a major mediator of the inflammatory cascade that underlies SJIA	United States	Pascual V et al., 2005 [95]
Analysis of serum from 39 patients with active AOSD, 32 with RA, and 40 HCs	Serum CXCL10 and CXCL13 levels may serve as clinical markers for assessment of disease activity, especially skin manifestations, in AOSD	South Korea	Han JH et al., 2015 [96]
Analysis of skin biopsy materials of 40 patients with AOSD, 10 with eczema, 10 with psoriasis, and 10 HCs	CXCR4 could be a clinical biomarker of evaluation for disease activity in AOSD. CXCR4/CXCL12 may influence skin manifestations of AOSD.	South Korea	Han JH et al., 2019 [97]

SJIA, systemic juvenile idiopathic arthritis; AOSD, adult–onset Still’s disease; HC, healthy control; RA, rheumatoid arthritis, ICAM–1, intracellular adhesion molecule–1; IL, interleukin; SLE, systemic lupus erythematosus.

**Table 2 ijms-22-13038-t002:** List of studies indicating association between NETs and SJIA/AOSD.

Study Population	Main Findings	Country	Author, Year [ref.]
Analysis of serum from 60 patients with SJIA, 148 with other inflammatory disease, and 50 HCs	S100A9/S100A9 and IL–1*β* represent a novel positive feedback mechanism activating phagocytes during the pathogenesis of systemic-onset JIA.	Germany	Frosch M et al., 2009 [104]
Analysis of serum from 20 patients with AOSD and 20 HCs	S100A8/A9 may be involved in the inflammatory response with induction of proinflammatory cytokines and may serve as a clinicopathological marker for disease activity in AOSD.	South Korea	Kim HA et al., 2016 [105]
Analysis of serum from 240 patients (60 with SJIA, 17 with FMF, 18 with neonatal–onset multisystem inflammatory disease, 17 with Muckle–Wells syndrome, 45 with leukemia, 83 with systemic infection), and 45 HCs	S100A12, a marker of granulocyte activation, is highly overexpressed in patients with systemic-onset JIA or FMF.	Germany	Wittkowski H et al., 2008 [106]
Analysis of serum from 37 patients with SLE and 38 HCs	S100A12 levels showed strong correlations with known disease activity markers.	South Korea	Bae C-B et al., 2014 [107]
Analysis of serum from 35 patients with AOSD and 20 HCs	Serum levels of cell-free DNA, myeloperoxidase-DNA complex, and α-defensin were significantly increased in patients with AOSD compared to HCs.	South Korea	Ahn MH et al., 2019 [27]
Analysis of serum from 37 SJIA patients without treatment, 32 with SJIA on treatment, and 16 HCs	Levels of neutrophil granulocytes in serum reflect underlying disease activities of JIA.	Sweden	Backlund M et al., 2021 [108]
Analysis of 109 patients with AOSD (active 78 and inactive 31), 29 with SLE, 29 with RA, and 62 HCs	Neutrophils-derived lipocalin–22 is higher in plasma and liver tissue in AOSD patients than in healthy controls.	China	Jia J et al., 2021 [109]
Analysis of serum from 38 patients with AOSD and 26 HCs	Fine-tuned mechanism between inflammatory (IL–18 induced NETs) and anti-inflammatory (microRNA–223) factors in AOSD	Taiwan	Liao T-L et al., 2021 [110]
Analysis of 30 patients with AOSD	LDGs and NETs (HMGB–1 and LL–37) are increased in patients with active AOSD and correlate with cutaneous manifestations, arthritis and fever.	Mexico	Torres-Ruiz J et al., 2019 [111]
Analysis of serum from 40 patients with AOSD and 40 HCs	Serum HMGB–1 levels were elevated in AOSD patients compared to the HCs and were correlated with CRP and systemic score.	South Korea	Jung J-Y et al., 2016 [112]
Analysis of serum from 12 patients with SJIA and 28 with other JIA	Serum HMGB–1 can be associated with clinical disease activity in JIA and is particularly at the highest level at the time of diagnosis.	Taiwan	Xu D et al., 2021 [113]
Analysis of serum and synovial fluids from 99 patients with SJIA, 19 with SLE, and 27 HCs	HMGB–1 and its soluble receptor RAGE in the blood and SF indicate that inflammation is triggered by alarmins in SJIA and SLE	Croatia	Bobek D et al., 2014 [114]
Analysis of serum from 66 patients with AOSD and 40 HCs	Demonstrated a close association between the increased levels of circulating NETs and organic involvement, as well as glucocorticoid responses in AOSD patients.	China	Jia Jin et al., 2020 [115]
Analysis of genotype from 164 patients with AOSD and 305 HCs	Functional *LILRA3* is a novel genetic risk factor for the development of AOSD.	China	Wang M et al., 2021 [116]
Analysis of serum from 26 patients with SJIA	The level of serum histones extracted from NETs released by activated neutrophils has a positive correlation with the activity of SJIA.	China	Hu X et al., 2019 [117]

NET, neutrophil extracellular trap; SJIA, systemic juvenile idiopathic arthritis; AOSD, adult–onset Still’s disease; HC, healthy control; IL, interleukin; FMF, familial Mediterranean fever; DNA, deoxyribonucleic acid; SLE, systemic lupus erythematosus; RA, rheumatoid arthritis; RNA, ribonucleic acid; LDG, low–density granulocytes; HMGB–1, high mobility group box–1; RAGE, receptor for advanced glycation end products; CRP, C–reactive protein.

## Data Availability

All available data are reported in the manuscript.

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
