# Peer review of "An Update on the Pathogenic Role of Neutrophils in Systemic Juvenile Idiopathic Arthritis and Adult-Onset Still’s Disease"

_ijms, 2021, doi:10.3390/ijms222313038_

Round 1

Reviewer 1 Report

The manuscript by Kim et al. is a very well written, well structured and interesting review article. I only have a few very minor comments.

On page 2 section 2. They list mediators induced by PAMP or DMAP signaling and there including C5a. C5a is not induced by this mechanism but by complement activation.

A few lines further down on the same page the authors say - , a lipid mediator class switching occur, . In my mind this made me first think of immunoglobulin class switching and it took me some time to understand that this was not the meaning. Possibly rephrase by saying something like this, a switch in lipid mediator type occurs,.

On the last page of conclusion 4 of 24 there is a repetition in the beginning of that page,

 ¨ undoubtedly, further research in this regard will provide greater clarity……….

Author Response

We appreciate your review of our manuscript “An update on the pathogenic role of neutrophils in systemic juvenile idiopathic arthritis and adult-onset Still’s disease”. In response to your comments and those of the reviewers, we have made several changes to the text, as summarized below.

Reviewer: 1
Thank you for your valuable comment.

The manuscript by Kim et al. is a very well written, well structured and interesting review article. I only have a few very minor comments.

  1. On page 2 section 2. They list mediators induced by PAMP or DMAP signaling and there including C5a. C5a is not induced by this mechanism but by complement activation.

Answer> Thank you for your comment. The part you pointed out was changed as follows, and the revised sentence was marked on the text.

  • In the early stages of inflammation, the initiating stimulus is sensed by pathogen-associated molecular patterns (PAMPs) or damage-associated molecular patterns (DAMPs), leading to the release of pro-inflammatory mediators, including lipid mediators such as prostaglandins, leukotriene B4, and complement anaphylatoxins C3a and C5a along with many cytokines and chemokines (e.g., tumor necrosis factor [TNF]-α, interleukin-1 [IL-1], and CXC chemokines).

  1. A few lines further down on the same page the authors say - , a lipid mediator class switching occur, . In my mind this made me first think of immunoglobulin class switching and it took me some time to understand that this was not the meaning. Possibly rephrase by saying something like this, a switch in lipid mediator type occurs,.

Answer> Thank you for your comment. As your comment, we changed the sentence as follows.

  • Simultaneously, with the recruitment of these phagocytic cells, a switch in lipid mediator type occurs, resulting in the biosynthesis of pro-inflammatory lipid mediators into anti-inflammatory lipid mediators, such as lipoxins, resolvins, and protectins, to attenuate inflammation.

  1. On the last page of conclusion 4 of 24 there is a repetition in the beginning of that page,

 ¨ undoubtedly, further research in this regard will provide greater clarity……….

Answer> Thank you for your comment. The repeated part was deleted and rewritten as follows.

  • Despite the continuous research conducted so far on this subject, undoubtedly, further research in this regard will provide greater clarity in understanding the pathogenesis of AOSD and SJIA.

Reviewer 2 Report

This manuscript provides a comprehensive review of recent findings of the roles of neutrophils and NET in the pathogenesis auto-immune diseases and their potentials in clinical applications such as biomarkers and disease monitoring. The information shall be of interest to the audiences targeted by this Journal. 

The manuscript is well written and the information presented in a thorough and logical manor. The figures are concise and clear, although the fonts could be enlarged for easier reading.

Author Response

We appreciate your review of our manuscript “An update on the pathogenic role of neutrophils in systemic juvenile idiopathic arthritis and adult-onset Still’s disease”. In response to your comments and those of the reviewers, we have made several changes to the text, as summarized below.

Thank you for your valuable comment.

This manuscript provides a comprehensive review of recent findings of the roles of neutrophils and NET in the pathogenesis auto-immune diseases and their potentials in clinical applications such as biomarkers and disease monitoring. The information shall be of interest to the audiences targeted by this Journal. 

The manuscript is well written and the information presented in a thorough and logical manor. The figures are concise and clear, although the fonts could be enlarged for easier reading.

Answer> Thank you for your comment. As your comment, the font size has been increased, and the image quality was also improved.